



**Seasonal development of iron limitation in the sub-Antarctic zone**
Thomas J. Ryan-Keogh[1,2], Sandy J. Thomalla[1,2], Thato N. Mtshali[1], Natasha R. van Horsten[1,3],
Hazel Little[2]
[1]Southern Ocean Carbon and Climate Observatory, Natural Resources and Environment,
CSIR, Rosebank, Cape Town, 7700, South Africa
[2]Department of Oceanography, University of Cape Town, Rondebosch, Cape Town, 7701,
South Africa
[3]Department of Earth Sciences, Stellenbosch University, Stellenbosch, 7600, South Africa
*Correspondence to*: tryankeogh@csir.co.za
**Abstract**
The seasonal and sub-seasonal dynamics of iron availability within the sub-Antarctic zone
(SAZ, ~40 – 45°S) play an important role in the distribution, biomass and productivity of the
phytoplankton community. The variability in iron availability is due to an interplay between
winter entrainment, diapycnal diffusion, storm-driven entrainment, iron scavenging and iron
recycling processes. Biological observations utilising grow-out iron addition incubation
experiments were performed at different stages of the seasonal cycle within the SAZ to
determine the importance of these supply mechanisms. Here we demonstrate that at the
beginning of the growing season there is sufficient iron to meet the demands of the
phytoplankton community, but as the growing season develops the supply mechanisms fail to
meet this demand. Phytoplankton increase their photosynthetic efficiency and net growth rates



following iron addition from mid to late summer, with no differences determined during early
summer; suggestive of seasonal iron depletion and low iron resupply. The result of which is
residual macronutrients at the end of the growing season, and the prevalence of the high-
nutrient low-chlorophyll (HNLC) condition. We conclude that despite the prolonged growing
season characteristic of the SAZ, which can extend into late summer/early autumn, the results
suggest that the iron supply mechanisms are insufficient to maintain potential maximal growth
and productivity throughout the season.



## 1. Introduction

The Southern Ocean is an important region for atmospheric $CO_2$ drawdown, 30-40% of global anthropogenic carbon uptake (Khatiwala et al., 2009; Mikaloff Fletcher et al., 2006; Schlitzer, 2002), which is driven by phytoplankton community production and the biological carbon pump (BCP). The BCP is however sensitive to environmental influences that are associated with climate change, which include an intensification of the westerly winds (Le Quéré et al., 2009), and altered upwelling and mixed layer stratification (Bopp et al., 2005; Boyd, 2002). Together, these changes will impact the light and nutrient supply to the phytoplankton community, which could in turn alter the efficiency and extent of the BCP in the future.

The high productivity characteristic of this region is driven in part by the high macronutrient availability, while phytoplankton growth and productivity is ultimately constrained by the availability of light and iron (de Baar et al., 1990; Martin et al., 1990). The result of this limitation is the prevalence of macronutrients in the surface waters at the end of the growing season, resulting in the paradoxical high nutrient low chlorophyll (HNLC) conditions characteristic of the region. Further controls on the seasonal evolution and extent of the phytoplankton bloom include potential silicate limitation (Boyd et al., 2010; Hutchins et al., 2001), top-down controls by meso- and micro-zooplankton grazing (Dubischar and Bathmann, 1997; Moore et al., 2013; Pakhomov and Froneman, 2004; Smetacek et al., 2004) and seasonal/sub-seasonal changes in the critical and mixed layer depths (Fauchereau et al., 2011; Nelson and Smith, 1991).

Iron is a key component of photosynthesis due to the high requirements in the formation and function of key photosynthetic proteins, including photosystems I and photosystem II (Raven, 1990; Shi et al., 2007; Strzepek and Harrison, 2004). In addition, iron requirements by phytoplankton are closely linked to light availability, displaying an inverse relationship. Under

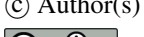



low light conditions phytoplankton can maximise photosynthesis in different ways; by either
increasing the size of their photosynthetic units or by increasing the number of their
photosynthetic units, the latter resulting in an increase in the iron requirements under low light
(Maldonado et al., 1999; Raven, 1990; Strzepek et al., 2012; Strzepek et al., 2011; Sunda and
Huntsman, 1997). This close coupling of light and iron that increases the cellular demand for
iron under low light can diminish light dependent photosynthesis when iron concentrations are
too low to support growth (Hiscock et al., 2008; Moore et al., 2013; Ryan-Keogh et al., 2017b).
Iron is also required in the function of both nitrate and nitrite reductase (de Baar et al., 2005),
which function to facilitate the assimilation of nitrate and nitrite and their subsequent
intracellular reduction to ammonium. In the Southern Ocean, and other HNLC areas, nitrate
uptake rates are reported as becoming iron limited for this reason (Cochlan, 2008; Lucas et al.,
2007; Moore et al., 2013; Price et al., 1994). However, rather than iron limitation directly
inhibiting nitrate/nitrite reductase activity the cause of reduced uptake rates may be the result
of a bottleneck further downstream due to a lack of photosynthetically derived reductant
(Milligan and Harrison, 2000). The result of this is the excretion of excess nitrate and nitrite
back into the water column, culminating in the HNLC condition that prevails in the Southern
Ocean.
The Atlantic sector of the Southern Ocean is composed of a series of water masses,
each with distinct physical and chemical properties, that are constrained by circumpolar fronts
with large geostrophic velocities (Nowlin and Klinck, 1986; Orsi et al., 1995). The differing
physical and chemical properties create a high degree of zonal variability within the biology,
in particular the timing and extent of phytoplankton seasonal blooms (Thomalla et al., 2011).
Key physical controls on this variability include sea ice cover and day length, yet this is not
enough to explain the full range of variability measured. An alternative approach has examined
whether the supply mechanisms of iron to the mixed layer differ significantly in their extent



allowing regions like the sub-Antarctic zone (SAZ) to exhibit prolonged summer blooms in
comparison to the polar front zone (PFZ) (Thomalla et al., 2011). Tagliabue et al. (2014)
postulated that due to weak diapycnal inputs of iron there must be a heavy reliance of Fe-
recycling within the mixed layer to meet the iron demand. An alternative hypothesis is that
summer eddy-storm interactions sustain mixed layer biomass through entrainment, particularly
in the SAZ (Carranza and Gille, 2015; Nicholson et al., 2016; Swart et al., 2015). As a storm
passes through the SAZ it deepens the mixed layer accessing the subsurface iron reservoir, the
subsequent re-shoaling of this buoyant water fuels surface water phytoplankton growth in a
high light environment. The drivers of the seasonal characteristics of these regions is likely a
combination of both factors with variable dominance in time and space. Regardless, a greater
understanding of the iron supply mechanisms and whether they meet the demand for
phytoplankton growth is required.

This paper aims to test whether the phytoplankton community in the sub-Antarctic zone

is seasonally limited by iron availability. This was done through a series of ship-board grow-
out nutrient addition incubation experiments were performed to determine the extent to which
the addition of iron at different times of the growing season would relieve the phytoplankton
from iron limitation driving an increase photosynthetic efficiency, biomass and growth
potential.

**2.   Materials and Methods**

**2.1. Oceanographic Sampling**

The samples and data presented here were obtained during the annual Austral summer relief
voyage of the South African National Antarctic Expedition 55 (SANAE 55) onboard the S.A.



Agulhas II to the Atlantic sector of the Southern Ocean as part of the Southern Ocean Seasonal
Cycle Experiment III (SOSCEx III, (Swart et al., 2012)); from the 3rd December 2015 to 11th
February 2016. During the cruise, 3 long-term (144 - 168 h) nutrient addition incubation
experiments were performed within the sub-Antarctic zone of the Atlantic sector of the
Southern Ocean (Fig. 1) to determine whether relief from iron limitation drove changes in
phytoplankton photophysiology and biomass (Table ). Uncontaminated whole seawater was
collected from 30 -  35 m depth in Teflon-lined, external closure 12 L Go-Flo samplers
deployed on a trace metal clean CTD rosette system.

**2.2. Nutrient addition incubation experiments**

Nutrient addition incubation experiments were performed using methods similar to those
employed previously in the Southern Ocean (Moore et al., 2007; Nielsdóttir et al., 2012; Ryan-
Keogh et al., 2017a) and the high latitude North Atlantic (Ryan-Keogh et al., 2013). Water for
experiments were allowed to settle in the Go-Flo samplers before being transferred unscreened
into acid-washed 50 L LDPE carboy (Thermo scientific) to ensure homogenization; the
homogenized water was then redistributed unscreened into 2.4 L polycarbonate bottles
(Nalgene) for the experiments. The triplicate initial samples were collected from the same 50L
LDPE carboy. Experiments during the cruise were incubated under two treatments, control and
iron addition (2.0 nM FeCl$_3$, 'Fe'), at a constant screened (LEE filters) light level of 129.45
µmol quanta m$^{-2}$ s$^{-1}$. Light levels were determined using a handheld $4\pi$ PAR sensor
(Biospherical Instruments), and were set on a day:night cycle according to the *in situ*
sunset/sunrise times. All experimental incubations were conducted as biological duplicates or
triplicates. Temperature was set at the in situ collection temperature for all samples. All bottle
tops were externally sealed with film (Parafilm), and bottles were double bagged with clear



polyethylene bags to minimize risks of contamination during the incubation. Subsampling of
all experiments occurred at the same time of day as the initial set-up.

**2.3. Chlorophyll a and Nutrient Analysis**

Samples for chlorophyll-a (Chl) analysis, 250 mL, were filtered onto GF/F filters and then
extracted into 90% acetone for 24 h in the dark at -20°C, followed by analysis with a
fluorometer (TD70; Turner Designs) (Welschmeyer, 1994). Macronutrient samples were
drawn into 50 mL diluvials and stored at -20°C until analysis on land. Nitrate + Nitrite and
Silicate were measured using a Lachat Flow Injection Analyser (Egan, 2008; Wolters, 2002),
whilst Nitrite and Phosphate were determined manually by colorimetric method as specified
by Grasshoff et al. (1983). Dissolved iron samples (DFe) were carefully collected in acid-
washed 125 mL LDPE bottles, acidified with 30% HCl suprapur to pH ~1.7 (using 2 mL L$^{-1}$
criteria) and stored at room temperature until analysis on land at UniBrest in France using the
Chemiluminescence – Flow Injection Analyser (CL-FIA) method (Obata et al., 1993; Sarthou
et al., 2003). Accuracy and precision of the method was verified by analysis of in-house internal
standards and SAFe reference seawater samples (Johnson et al., 2007); the limits of detection
were in order of 10 pM.

**2.4. Phytoplankton Photosynthetic Physiology**

Variable chlorophyll fluorescence was measured using a Chelsea Scientific Instruments
FastOcean fast repetition rate fluorometer (FRRf) integrated with a FastAct laboratory system.
Samples were acclimated in dark bottles at *in situ* temperatures, and FRRf measurements were
blank corrected using carefully prepared 0.2 µm filtrates for all samples (Cullen and Davis,





2003). Protocols for FRRf measurements consisted of the following: $100 \times 2$ µs saturation
flashlets with a 2 µs interval, followed by $25 \times 1$ µs relaxation flashlets with an interval of 84
µs with a sequence interval of 100 ms. Sequences were repeated 32 times resulting in an
acquisition length of 3.2 s. The power of the excitation LED ($\lambda450$) was adjusted between
samples to saturate the observed fluorescence transients within a given range of $R\sigma_{PSII}$ (the
probability of a reaction centre being closed during the first flashlet). $R\sigma_{PSII}$ was optimised
between 0.042 to 0.064 as per the manufacturer's specifications. By adopting this approach, it
ensures the best signal-to-noise ratio in the recovered parameters whilst accommodating
significant variations in the photophysiology of the phytoplankton community without having
to adjust the protocol. Data from the FRRf were analysed to derive the fluorescence parameters
as defined in Roháček (2002), by fitting transients to the model of Kolber et al. (1998).

**2.5. Phytoplankton Composition**

Pigment samples from the incubation experiments were collected by filtering $0.5 - 2.0$ L of
water onto 25 mm GF/F filters. Filters were frozen and stored at -80°C until analysis in
Villefranche, France on a HPLC Agilent Technologies 1200. Filters were extracted in 100%
methanol, disrupted by sonication, clarified by filtration and analysed by HPLC following the
methods of Ras et al. (2008); limits of detection were on the order of 0.1 ng L$^{-1}$. Pigment
composition data were standardized through root square transformation before cluster analysis
utilizing multi-dimensional scaling, where similar samples appear together and dissimilar
samples do not. Samples were grouped and analysed in CHEMTAX (Mackey et al., 1996)
using the pigment ratios from Gibberd et al. (2013). Multiple iterations of pigment ratios were
used to reduce uncertainty in the taxonomic abundance as described in (Gibberd et al., 2013),
with the solution that had the smallest residual used for the estimated taxonomic abundance.





### 2.6. Ancillary physical data


Temperature and salinity profiles were obtained from a Sea-Bird CTD mounted on the rosette
system. The mixed layer depth was calculated following de Boyer Montégut et al. (2004),
where the temperature differs from the temperature at 10 m by more than 0.2°C ($\Delta T_{10m} =$
0.2°C). The position of the fronts were determined using sea surface height (SSH) data from
maps of absolute dynamic topography (MADT) (Swart et al., 2010). The percentage euphotic
depth was calculated as a function of the natural log of *in situ* photosynthetically active
radiation (PAR) and the diffuse attenuation coefficient $K_z$.

### 2.7. Glider Dataset


Autonomous Seagliders (SG542 & SG543) were deployed in mooring mode in the sub-
Antarctic zone of the Southern Ocean (43°S 8.5°E) as part of SOSCEx III. SG543 was
deployed from 28 July 2015 to 8 December 2015, followed by SG542 which continued
sampling until 8 February 2016. The deployment of both gliders resulted in a continuous high-
resolution time series of 1832 profiles over 196 days, down to depths of 1000 m. The gliders
measured a suite of parameters including conductivity, temperature, pressure, PAR,
fluorescence and optical backscattering at two wavelengths ($\lambda = 470$ and $700$). At the
deployment and retrieval of each glider cross-calibration CTD casts were performed (all within
3km and 4 h of each other), yielding independent inter-calibrations between glider sensors and
bottle samples of Chl. Glider fluorescence was corrected for quenching and converted to units
of Chl (mg m$^{-3}$) as described in Thomalla et al. (2017). Wind stress (N m$^{-2}$) data was collected



from a weather station mounted on a simultaneous deployment of a Liquid Robotics Wave
Glider; wind stress was corrected to 10 m using the wind profile power-law (Irwin, 1967).

**2.8. Data analysis**

Sample means and standard deviations were calculated using Python, followed by tests for
normality and equal variance prior to analysis of variance to determine treatment effects (SciPy
v0.17.1, Python v3.6). Significant results are reported at the 95% confidence level ($p < 0.05$).

**3. Results**

The experiment set-up location in the SAZ spanned 66 days from the initiation of the first
experiment to the initiation of the third experiment. Chlorophyll concentrations did not vary
substantially between initiations, ranging from $0.84 - 0.97$ µg L$^{-1}$, alongside no significant
variations in temperature and salinity (Table 1). Silicate concentrations remained fairly
constant between experiments, with concomitant decreases in phosphate and dissolved iron
(DFe) concentrations; whereas nitrate concentrations increased throughout the growing season.
Photophysiological measurements of quantum efficiency ($F_v/F_m$) ranged from $0.19 - 0.30$ with
decreases in the cross-section of PSII ($\sigma_{PSII}$) from 14.79 to 7.08 nm$^{-2}$. All experiments were set
up with water collected from above the mixed layer and the mean euphotic depth of
$63.89 \pm 19.13$ m, with the percentage of surface light ranging from $14.83 - 10.66\%$.

Data from 144-168 h experiments in the SAZ indicated variable responses to iron addition

to the extant phytoplankton community (Fig. 2). During 'early-summer' (experiment 1), no
evidence for iron stress was observed as indicated in the similar responses in $F_v/F_m$ (Fig. 2a)
and chlorophyll (Fig. 2b) between iron addition (+ Fe) and control treatments; both variables

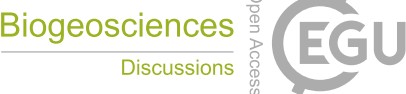

increased to similar values at the end time point. Statistical analysis confirmed that there were
no significant differences in $F_v/F_m$ or chlorophyll. The effective cross-section of PSII ($\sigma_{PSII}$
($nm^{-2}$)) displayed a similar pattern with no significant differences between treatments,
decreasing in both treatments to $5.68\pm0.27$ and $5.63\pm0.13$ for the control and iron addition
treatments respectively. Experiment 2, initiated 28 days later in 'mid-summer', did exhibit
signs of potential iron limitation (Fig. 2c, 2d). $F_v/F_m$ increased from $0.30\pm0.02$ to maximum of
$0.39\pm0.01$ at 120 h, whilst the control treatment ranged between 0.27 and 0.34 (Fig. 2c).
Moreover, chlorophyll concentrations were over 2 times higher in the iron addition treatment
compared to the controls at the end time point (Fig. 2d). Significant differences were observed
for $F_v/F_m$ from 72 h onwards and for chlorophyll concentrations from 120 h onwards. $\sigma_{PSII}$
decreased to a minimum of $4.62\pm0.15$ $nm^{-2}$ in the iron addition at 120 h, corresponding to the
highest value in $F_v/F_m$; whereas the control treatment decreased from $6.45\pm0.23$ to $5.96\pm0.13$
$nm^{-2}$. The final experiment in 'late-summer' (experiment 3) displayed similar evidence for
potential iron limitation within the extant phytoplankton community (Fig. 2e, f). $F_v/F_m$ in the
control treatment remained constant at $0.26\pm0.01$, whereas in the iron addition treatment it
increased to $0.33\pm0.01$ (Fig. 2e). Chlorophyll concentrations were 2.5 times higher in the iron
addition treatment compared to the controls after 144 h (Fig. 2f), resulting in significant
differences in $F_v/F_m$ from 24 h and in chlorophyll concentrations at the end time point. $\sigma_{PSII}$
also decreased to a greater extent than the control in experiment 3 from $7.08\pm0.48$ $nm^{-2}$ to a
minimum of $5.45\pm0.15$ $nm^{-2}$, compared to $6.23\pm0.14$ $nm^{-2}$ in the control at 144 h.

Chlorophyll specific growth rates ($\mu^{Chl}$) were calculated for each experiment (Table 2,

Supplementary Information Fig. S1), displaying significantly higher growth rates for the iron
addition treatment in experiments 2 and 3 by up to 50% and 63% respectively, with no
significant differences in experiment 1. Enhanced nitrate drawdown $\Delta(NO_3^-)$ was exhibited in
experiment 2 (Table 2), with rates approximately 4 times higher than the other experiments.





No enhanced drawdown of phosphate or silicate was exhibited in any of the experiments.
Taxonomic abundance (Supplementary Information, Fig. S2), indicated that the dominant
component of the community was Haptophytes (>40%) when all experiments were initiated.
Experiment 1 displayed significant increases in Diatoms in both treatments, alongside a
significant increase in Synechococcus in the control treatment. Experiments 2 and 3 displayed
similar results with significant increases in Diatoms following iron addition, with reductions
in the Haptophyte group.

$F_v/F_m$ is derived from measurements and analysis of the fluorescence kinetics of the

photosynthetic reaction centre photosystem II (PSII) and associated light-harvesting antenna
proteins (Kolber and Falkowski, 1993). Understanding the mechanistic changes in $F_v/F_m$ can
provide information on how the phytoplankton community respond to different stress factors.
Increases in $F_v/F_m$ following iron enrichment do not appear to be the result of an increase in
PSII efficiency ($F_v$), but rather due to decreases in $F_m$ and $F_o$ (Behrenfeld et al., 2006; Lin et
al., 2016; Macey et al., 2014; Ryan-Keogh et al., 2017a). To determine these relative changes
in photophysiology, the absolute difference in $F_v/F_m$ between the control and iron addition
bottles was calculated at 24 h, $\Delta(F_v/F_m)$ (Ryan-Keogh et al., 2013). $\Delta(F_v/F_m)$ in experiment 1
was indistinguishable from zero (Fig. 3a), whereas in experiment 2 and 3 it was consistently
positive with values of $0.08 \pm 0.01$ and $0.06 \pm 0.00$ respectively. These responses were markedly
similar to the absolute differences in growth rates (Fig. 3b), with significantly higher
differences in experiments 2 and 3. The absolute changes in maximum fluorescence ($F_m$, Fig.
3c) and variable fluorescence ($F_v$, Fig. 3d) normalized to chlorophyll were calculated to
determine the mechanistic response. Significant differences were determined for $F_m$ Chl$^{-1}$ in
experiments 2 and 3, with no significant differences in $F_v$ Chl$^{-1}$ across any experiments.

**4.  Discussion**

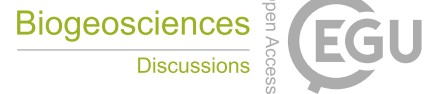




Photosynthesis in the Southern Ocean is considered to be limited in winter by low mean
irradiance, with net phytoplankton growth rates increasing rapidly following the onset of
stratification in spring (Sverdrup, 1953). Despite these high levels of productivity and growth,
complete macronutrient drawdown is not possible due primarily to constraints in the
availability of iron (Boyd et al., 2007; de Baar et al., 1990). Reasons for this growth limitation
include the high iron requirements of the photosynthetic apparatus (Raven et al., 1999; Shi et
al., 2007) particularly under low light conditions and a lack of iron sources (Duce and Tindale,
1991; Tagliabue et al., 2014). Phytoplankton blooms in the SAZ are characterized by high
inter-annual and intra-seasonal variability with an extended duration that sustains high
chlorophyll concentrations late into summer (Carranza and Gille, 2015; Swart et al., 2015;
Thomalla et al., 2011; Thomalla et al., 2015). The longevity of these blooms is unusual as Fe
limitation at this time of year is expected to be limiting growth (Boyd, 2002). To determine the
extent to which the availability of iron is restricting phytoplankton photosynthesis and biomass
accumulation in the SAZ, a series of grow-out nutrient addition incubation experiments were
performed during the austral summer of 2015/2016.

The nutrient addition experiments (Fig. 2) demonstrated the development of seasonal

iron limitation of the in situ phytoplankton population within the SAZ from early summer
(December) to late summer (February). Experiment 1, which was set up during the early
growing season did not display any significant differences between treatments with all
parameters changing comparatively between the iron addition and control (Fig. 2a, 2b). The
rapid increase in $F_v/F_m$ in both treatments at 24 h is likely due to potential bottle effects i.e. a
change in the light environment. Experiment 2 displayed the greatest response to the addition
of iron with significantly different $F_v/F_m$ (Fig. 2c), chlorophyll derived net growth rates (Fig.
S1) and nitrate drawdown rates (Table 2). Experiment 3 also continued this trend with



significantly higher $F_v/F_m$ (Fig. 2e) and net growth rates in the iron addition treatment
compared to the control (Fig. S1). The addition of iron also resulted in community level
changes switching from haptophyte dominated communities to diatom dominated communities
(Fig. S2). However, it remains to be determined whether this is shift towards small diatoms or
large diatoms, as large diatoms would require an increased silicate concentration which is a
limiting macronutrient in this region (Hutchins et al., 2001). Irregardless, this community shift
suggestive of community specific iron quota requirements (Ryan-Keogh et al., 2017a; Strzepek
et al., 2012; Strzepek et al., 2011), which drive the composition of the extant phytoplankton
community in the SAZ. The increased responses with time are indicative of seasonal iron
limitation, similar to the high latitude North Atlantic (Ryan-Keogh et al., 2013), where
potential iron sources are depleted early in the growing season resulting in HNLC conditions
at the end of the growing season.

Mechanistic changes in $F_v/F_m$, i.e. $\Delta(F_v/F_m)$, are a useful proxy to determine the

potential physiological signal of iron limitation without any superimposing taxonomic signal
(Suggett et al., 2009). The derived variable $\Delta(F_v/F_m)$ was higher in experiments in 2 and 3 (Fig.
3a), with values consistent with studies from the North and South Atlantic and the Ross Sea
(Browning et al., 2014; Ryan-Keogh et al., 2017a; Ryan-Keogh et al., 2013), which correlated
well with the observed differences in net growth rates ($\Delta\mu^{Chl}$, Fig. 3b). Whilst no empirical
relationship should be inferred between measures of photophysiology and measures of growth
rates (Kruskopf and Flynn, 2006; Parkhill et al., 2001; Price, 2005), the observed correlation
between these two independent variables suggest that a biomass independent measure of
physiological iron stress, $F_v/F_m$, is likely accompanied by a significant repression of
phytoplankton growth rates. These experiments also provide insights into the mechanistic iron-
stress response of phytoplankton photophysiology, where increases in $F_v/F_m$ following iron
addition are due to a reduction in the ratio of $F_m\,Chl^{-1}$ rather than $F_v\,Chl^{-1}$ (Fig. 3c, 3d). This in





agreement with similar observations made in the Ross Sea, the high latitude North Atlantic and
equatorial Pacific (Behrenfeld et al., 2006; Lin et al., 2016; Macey et al., 2014; Ryan-Keogh et
al., 2017a), all regions where the phytoplankton communities are subject to iron limitation.
Elevated ratios of $F_m$ $Chl^{-1}$ are potentially indicative of an energetically-decoupled pool of
chlorophyll that possess a higher fluorescence yield than PSII at $F_m$ (Macey et al., 2014; Ryan-
Keogh et al., 2017a; Ryan-Keogh et al., 2012; Schrader et al., 2011). These pools can be
significant in iron limited regions with important implications for chlorophyll derived primary
productivity estimates that can be overestimated as a result (Behrenfeld et al., 2006; Macey et
al., 2014).

The seasonal development of iron stress in the SAZ is suggestive of a primary dominant

iron source to the surface waters, winter entrainment, which is subsequently depleted by upper
ocean biota and abiotic scavenging onto settling particles (Tagliabue et al., 2014). Although
diapycnal diffusion resupplies the mixed layer from late spring onwards, its low rates cannot
be reconciled with potential phytoplankton uptake (Tagliabue et al., 2014). Instead, Tagliabue
et al. (2014) propose that biologically recycled iron within the mixed layer is the dominant
mechanism for sustaining summertime blooms. However, there is now compelling evidence to
suggest that storm events may also play a critical role in extending the duration of summertime
production through intra-seasonal entrainment of dissolved iron from a subsurface reservoir
(Carranza and Gille, 2015; Fauchereau et al., 2011; Swart et al., 2015; Thomalla et al., 2011).
This mechanism was tested using a 1D biogeochemical model by Nicholson et al.
(2016) whose results suggest that intra-seasonal mixed layer perturbations may offer relief
from iron limitation in summer, particularly if there is sufficient subsurface vertical mixing
beneath the surface mixed layer.

A SAZ glider study by Little et al. (in review) corroborated these findings with summer

match ups in small-scale temporal variability (< 10 days) in wind stress, MLD and chlorophyll



that emphasise the interconnectedness between the physical drivers and their biological

response. Despite the similarity in the scales of variability, no correlation was observed

between MLD and chlorophyll, which is explained by the variable response that MLD

adjustments drive i.e. dilution (a decrease in chlorophyll with increasing MLD) and growth (an

increase in chlorophyll with increasing MLD in response to nutrient entrainment) (Fauchereau

et al., 2011). Both these scenarios can be observed in the glider time series from this study (Fig.

4), where increased wind stress and deeper MLD's were associated with both reduced (15 – 29

December) and enhanced (29 January – 7 February) chlorophyll. Both of the summer

experiments were set up during periods of low wind stress ($<0.2$ N m$^{-2}$) with shallow MLDs,

which explains the positive response to iron relief observed in experiments 2 and 3. Worth

noting is the time period between 10 January and 29 January where the SAZ experienced

uncharacteristically low winds for an extended period of time that drove very shallow MLD's

(~20 m) and the development of a subsurface chlorophyll bloom. During this period, mean

chlorophyll concentrations in the mixed layer (~0.4 mg m$^{-3}$) were lower than the euphotic zone

(~0.8 mg m$^{-3}$), indicative of an iron supply within the mixed layer that is not sufficient to meet

phytoplankton demands (i.e. surface water iron recycling is insufficient to sustain summertime

productivity).

The decreasing DFe concentrations from the experimental depths do appear to suggest

that DFe may not be sufficient, but this may not be a good indicator of iron stress as any limiting

nutrient would be expected to be severely depleted through biological uptake with a resultant

ambient concentrations that would remain close to zero despite possible event scale supply

(Ryan-Keogh et al., 2017a). Furthermore, precaution must be taken when investigating changes

in chlorophyll concentrations, as chlorophyll is only a proxy for phytoplankton biomass

(Behrenfeld et al., 2016; Bellacicco et al., 2016; Mignot et al., 2014; Westberry et al., 2008;



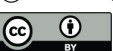

Westberry et al., 2016) and an increase in the average concentration over the euphotic zone
may represent a chlorophyll packaging effect due to lower light levels at depth.

The short transient periods of increased wind stress thus appear to provide temporary

relief from iron limitation, and when examined across the entire growing season can provide a
significant source of iron. Whilst the chlorophyll concentrations in late summer do not reach
the spring maximum, they remain approximately double the winter time average (~0.4 mg m$^{-}$
$^{3}$). However, the increase of in situ nitrate concentrations are suggestive of community level
iron limitation, as iron limitation can reduce the availability of photosynthetic reductant for
nitrate reduction which can lead to the excretion of excess nitrate back into the water column
(Cochlan, 2008; Lucas et al., 2007; Milligan and Harrison, 2000; Moore et al., 2013; Price et
al., 1994). Irrespective of the different supply mechanisms; winter-entrainment, storm driven
entrainment, diapycnal diffusion or microbial regeneration, the iron supply to the mixed layer
is not sufficient for phytoplankton primary production to completely drawdown all available
macronutrients. Moreover, this seasonal iron limitation may not be the only cause of sub-
maximal productivity rates as silicate can also potentially limit phytoplankton growth in this
region (Boyd et al., 2010; Hutchins et al., 2001). However, the significant shifts to diatom from
haptophyte communities (Fig. S2) within the experimental treatments following iron addition
suggest that silicate limitation may only be secondary limiting factor.

The current study represents an analysis of the seasonal development of iron limitation

in the SAZ, highlighting how a lack of sufficient iron supply relates to potential community
iron limitation. This is important for understanding Fe demand by the biota given the climate-
mediated variability in supply mechanisms (i.e. atmospheric deposition) (Mackie et al., 2008),
mixed layer depths and sea-ice cover (Boyd et al., 2012), as well as phytoplankton phenology
(Strzepek et al., 2012). The biogeochemical significance of the Southern Ocean, including the
highly productive Atlantic sector, will increase with respect to climate change (Marinov et al.,



2006); particularly as the Southern Ocean is the only HNLC region where the cryosphere is
critical to seasonal dynamics. Climate-mediated changes to iron supply will influence the
overall extent of phytoplankton growth, macronutrient drawdown and ultimately the strength
and efficiency of the biological carbon pump. However, the variations of supply in the seasonal
cycle will also continue to play an important role in this ecological important oceanic region
and warrant further investigation.

**Acknowledgements**

We would like to thank the South African National Antarctic Programme (SANAP) and the
captain and crew of the SA Agulhas II for their professional support through the cruise. We
would also like to thank the engineers and glider pilots from Sea Technology Services for
their professional support. Ryan Cloete and Ryan Miltz were involved in water collection and
experimental set up. This work was undertaken and supported through the CSIR's Southern
Ocean Carbon and Climate Observatory (SOCCO) Programme (http://socco.org.za/). This
work was supported by CSIR's Parliamentary Grant funding (SNA2011112600001) and the
NRF SANAP grant (SNA14073184298).



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





**Table 1: Locations of experiments conducted during the cruise along with details of the**
**initial set up conditions.**

| Experiment | Experiment 1 | Experiment 2 | Experiment 3 |
|---|---|---|---|
| | 'Early Summer' | 'Mid-Summer' | 'Late Summer' |
| Run time (h) | 168 | 168 | 144 |
| Initiation Date | 08/12/2015 | 05/01/2016 | 08/02/2016 |
| Latitude (°S) | -42.693 | -42.693 | -43.000 |
| Longitude (°E) | 8.738 | 8.737 | 8.500 |
| Collection Depth (m) | 30 | 35 | 35 |
| Sunrise:Sunset (GMT) | 03:30 – 18:30 | 04:00 – 19:00 | 04:40 – 18:40 |
| Chl-a (µg L$^{-1}$) | 0.97 | 0.84 | 0.90 |
| Nitrate (µM) | 10.60 | 12.80 | 18.25 |
| Silicate (µM) | 1.46 | 1.43 | 1.39 |
| Phosphate (µM) | 0.88 | 0.76 | 0.45 |
| DFe (nM) | 0.16 | 0.17 | 0.05 |
| $F_v/F_m$ | 0.19±0.06 | 0.30±0.02 | 0.26±0.01 |
| $\sigma_{PSII}$ (nm$^{-2}$) | 14.79±2.46 | 6.45±0.40 | 7.08±0.48 |
| MLD (m) | 33.77 | 56.96 | 43.32 |
| Salinity | 33.87 | 33.70 | 34.11 |
| Temp (°C) | 10.80 | 10.44 | 10.80 |
| % Light Depth | 14.83 | 11.59 | 10.66 |







**Table 2: Net growth rates calculated from chlorophyll accumulation ($\mu^{Chl}$) and nitrate**
**drawdown ($\Delta(NO_3^-)$) over the full experimental running time (t = 168, 168, 144 h).**
**Shown are averages with ± standard deviations, where n = 5.**

| Experiment | $\mu^{Chl}$ ($d^{-1}$) 0 - end | | $\Delta(NO_3^-)$ ($\mu$mol $L^{-1}$ $d^{-1}$) | |
|---|---|---|---|---|
| | + Fe | Control | + Fe | Control |
| 1 | 0.28±0.02 | 0.27±0.02 | 0.98±0.005 | 0.82±0.07 |
| 2 | 0.23±0.01 | 0.11±0.01 | 4.29±0.43 | 3.19±0.54 |
| 3 | 0.23±0.01 | 0.09±0.01 | 0.78±0.11 | 0.91±0.15 |






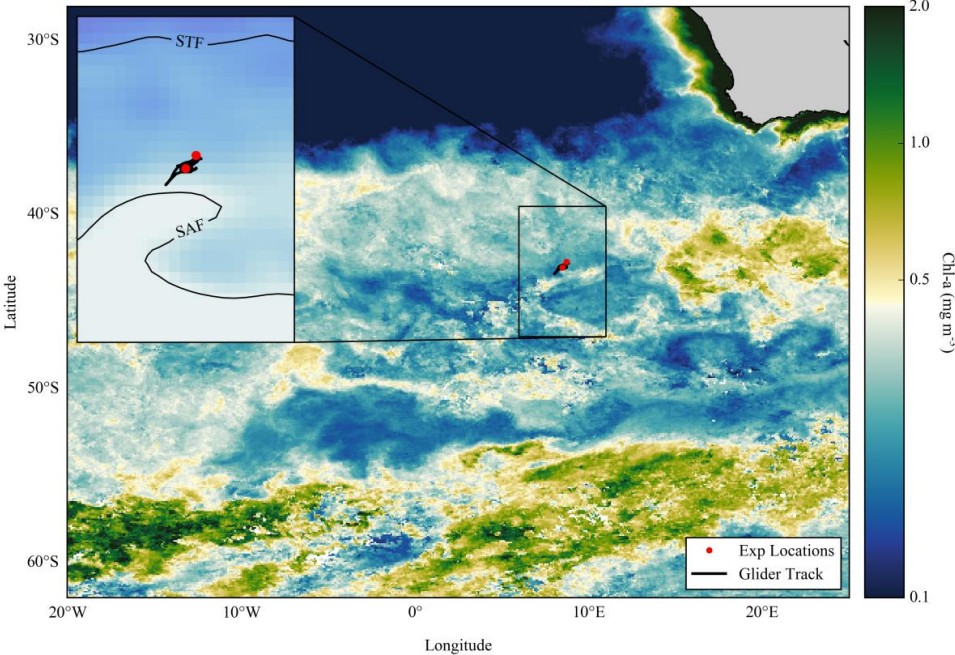

**Figure 1: Composite map of MODIS (8-day, 9 km) derived chlorophyll-a (mg m$^{-3}$) from December 2015 to February 2016 for the Atlantic sector of the Southern Ocean, with locations of nutrient addition incubation experiments and the glider track. Inset composite map of MADT from the CLS/AVISO product (Rio et al., 2011) from December 2015 to February 2016 with boundary definitions of sub-tropical front (STF) and sub-Antarctic front (SAF) (Swart et al., 2010), with locations of experiments and glider track.**



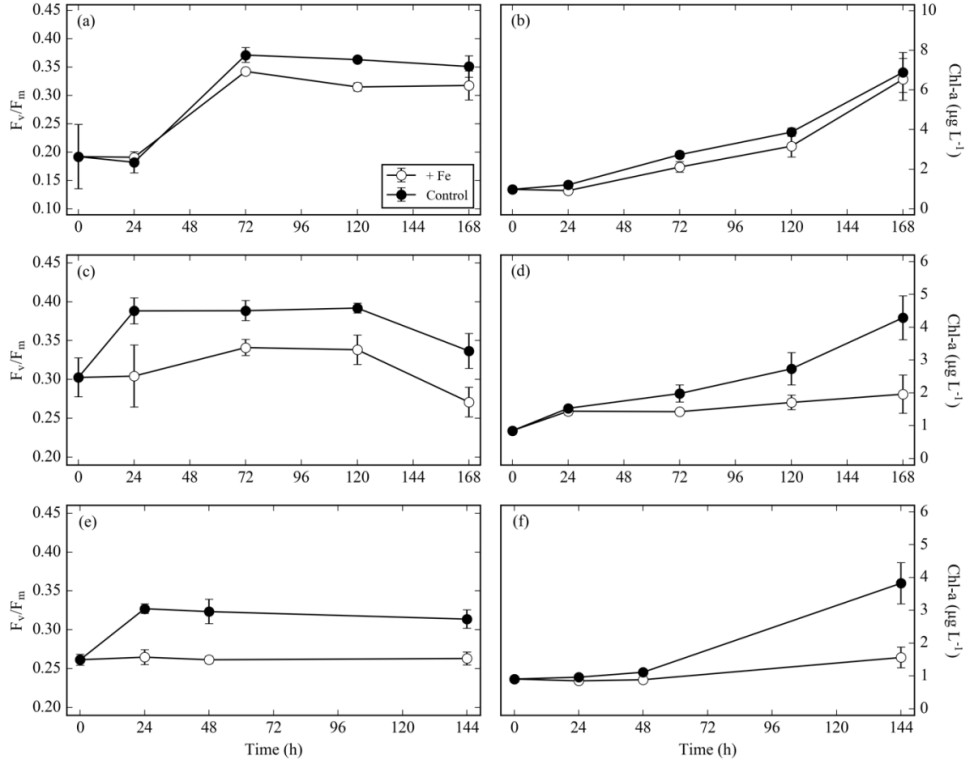

704

**Figure 2: $F_v/F_m$ (a, c, e) and chlorophyll-a (Chl-a) responses ($\mu g$ $L^{-1}$) (b, d, f), from the**

**control and Fe addition treatments of experiments initiated in the sub-Antarctic zone**

**over early summer (a, b), mid-summer (c, d), and late summer (e, f). Displayed here are**

**averages with ± standard deviations ($n = 3$ for all time points, except the end time point**

**where $n = 5$). Please note the different scales in panels a and b.**





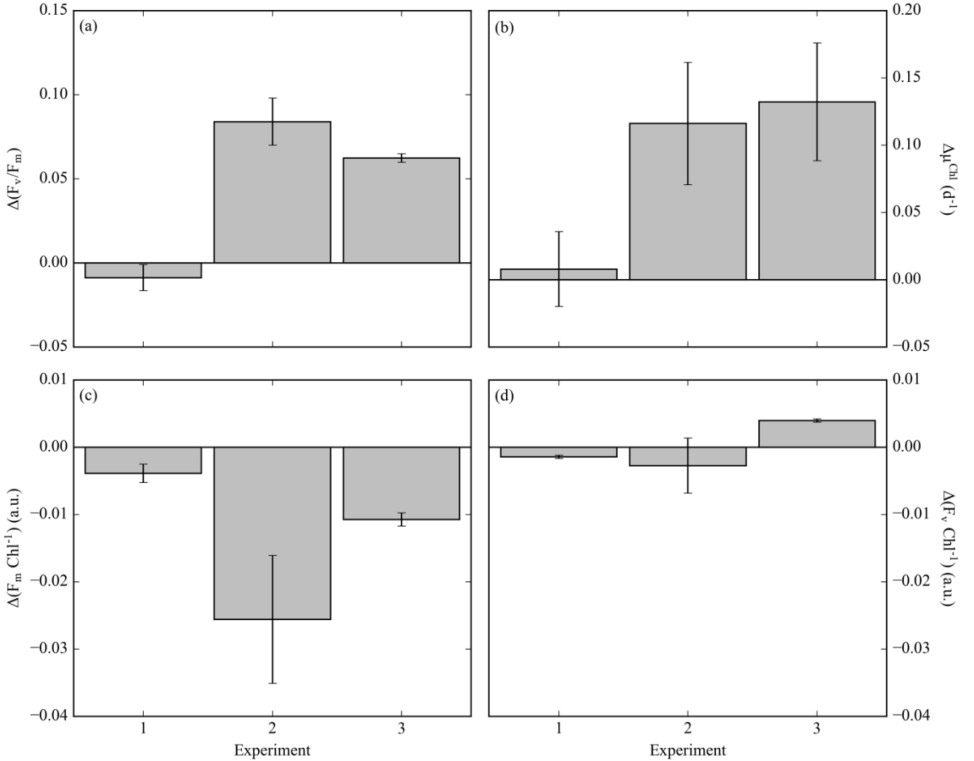

710

**Figure 3: (a) The difference in $F_v/F_m$ between the Fe treatment and control treatment**

**($\Delta(F_v/F_m)$) at the 24 h time point for experiments initiated in early summer (experiment**

**1), mid-summer (experiment 2) and late summer (experiment 3). (b) The difference in**

**chlorophyll derived net growth rates ($\Delta\mu^{Chl}$ ($d^{-1}$)), where t = 168, 168 and 144 h. (c) The**

**change in chlorophyll normalised maximum fluorescence, ($\Delta F_m$ $Chl^{-1}$). (d) The change in**

**chlorophyll normalised variable fluorescence, ($\Delta F_v$ $Chl^{-1}$). Displayed here are averages**

**with ± standard deviations ($n = 3 - 5$).**



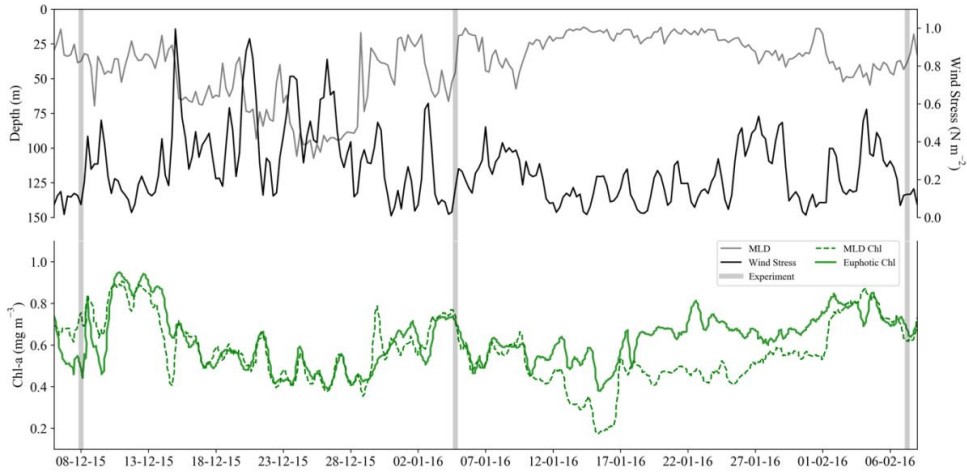


**Figure 4: Time series from 6th December 2015 to 8th February 2016 of (a) surface wind**


**stress (N m$^{-2}$), mixed layer depth (MLD, m) where $\Delta T_{10m} = 0.2°C$, and (b) mean**
**chlorophyll-a concentration (mg m$^{-3}$) from the MLD and the euphotic zone. Experiment**
**initiation dates are overlaid in grey bars.**