# Peer review of "Seasonal development of iron limitation in the sub-Antarctic zone"

_Biogeosciences, 2017_

## Referee Comment (RC1) · Anonymous Referee #1 · 19 Feb 2018

This manuscript presents an attempt to evaluate the seasonal cycle of iron stress in the sub-Antarctic zone using three bottle-scale iron enrichment experiments conducted in December–February. The novel aspect of the study is the reoccupation of the experimental site over 2 months. Whilst the these observations cannot, by some margin, be used to confidently state overarching changes in SAZ iron stress in these months, they are still valuable to the scientific community and worthy of publication in Biogeosciences. I however have a number of comments that should be addressed prior to publication. In particular I think the authors should more carefully/critically evaluate how far their experiments can actually be used to evaluate the seasonal development of Fe limitation in the sub-Antarctic zone without an analysis of supporting depth-resolved Fe, mixed layer depths, and PAR data. Upon reflection of the former, some rephras-
ing of the manuscript is required. Some additional important method details are also lacking.

The paper is in general well written and referenced and the figures and tables are clear and complete. My comments below are listed in order through the manuscript, not by importance.

Specific comments:

Line 18: Variability in iron supply also includes dust (not mentioned)

Line 21–22: 'incubation experiments were used to determine the importance of (iron) supply mechanisms'. Can they really be used for this? All they actually indicate is the biological response to incubation, not sources? A thorough analysis of Fe supply and demand, in conjunction with the bioassays, would be needed to do this (only 3 Fe values are reported).

Line 26–27: The results presented do not support the claim of progressive Fe depletion—phytoplankton appear to respond more to the Fe later in summer, but the in-situ Fe concentrations stay the same.

Line 72–73: Whilst excess nitrate+nitrite excretion under Fe stress may play a role, I think most would argue that high rates of resupply (relative to inefficient biological removal) overwhelmingly control the elevated residual nitrate in the Southern Ocean.

Line 97: 'were' change to 'that were'?

Line 99: Do the results actually indicate a change the photosynthetic efficiency? As the authors say themselves, most of the Fv/Fm change could well be due to pigment changes that have little to do with PSII efficiency.

Line 99: Change 'biomass' to 'chlorophyll-a biomass'?

Line 113: Table 'X'?

BGD
Line 115: CTD abbreviation defined?

Line 122: Why was water 'allowed to settle' in Go-Flo samplers? To increase the overall phytoplankton concentrations in the incubation bottles?

Section 2.2: - More details on the incubation experiment setup needed: What was the actual incubator? A culture cabinet? Something custom built? Please give details. - How does the PAR values supplied in the incubator differ from in-situ values? For this the authors will need to calculate an average ML PAR using their observations of ML, CTD PAR and ship-instrument PAR. Although this might be a pain, it might really help to pick apart the difference in growth environments experienced by the community just prior to incubation, and thereby help to interpret their response to the altered conditions.

Lines 130-131: Which experiments were in duplicates and which in triplicates? Figure 2 states n=3 or n=5, so I do not understand this. Please clearly indicate number of biological replicates (number of bottles with the same treatment) and technical replicates (i.e. FRR/chl measurements made from the same bottle).

Line 133-134: Can this approximate time of sub-sampling be included in Table 1? This might help to interpret the results (for instance, if daytime, the Fv/Fm increase in Exp 1 due to differences in PSII damage/down regulation between days).

Line 144: Exactly how were the dFe samples filtered (method, on-ship/land)?

Line 154: Was the instrument a FastTrackall?

Line 168: Was the FastPro software used or was custom code used? If the latter please give details.

Section 2.8: What was the test for means comparison between treatments? T-test?

Lines 303-304: 'The rapid increase in Fv/Fm in both treatments at 24 h is likely due to bottle effects i.e. a change in light environment.' - This experiment needs discussing
in more detail as the community response is clearly compatible with relief of resource limitation of larger cells. How did the light change between in-situ conditions and that in the incubator? What was the integrated PAR over the previous hours prior to the Fv/Fm measurement being made (if not at night time)? Perhaps different levels of PSII damage/down-regulation could be an explanation. Could the observed community shift not contribute to the Fv/Fm increases, i.e. do larger cells not typically have higher Fv/Fm (Sugget et al. 2009)? Should any chance of Fe contamination in the control bottles (e.g. during the 50L carboy) be acknowledged, as this would also be consistent with the observed responses?

References to seasonal Fe supply (e.g. lines 315-318, 341 onwards): Reduced Fe concentrations through the growing season are not actually observed. Furthermore, Fv/Fm values are low at both the beginning and end of the growing season. The only data the authors have to go on is the more pronounced phytoplankton responses to Fe relative to controls later in the growing season. Please rephrase these sections to more clearly indicate specifically what your bioassay results can actually say about seasonal changes in Fe supply to mixed layer waters.

Lines 311-312: 'large diatoms would require an increased silicate concentration, which is a limiting macronutrient in this region'. Is silicate a limiting nutrient at your site? Concentrations over 1uM were measured and chlorophyll-a and diatoms were enhanced in all +Fe treatments without added silicate.

Line 369-373: The authors state mean chlorophyll over the euphotic zone was higher than that over that of the mixed layer and then interpret this as a result of insufficient iron within the mixed layer. Whilst this is a possibility, could accumulation of lower light acclimated (higher chlorophyll/cell) phytoplankton below the mixed layer not equally play a role?

Lines 374–378: The mechanism the authors describe to explain the lack of Fe stress despite low Fe concentrations is not clear. The authors state that the 'limiting' nutrient
'would be expected to be severely depleted through biological uptake regardless of resupply'. But the authors state that they observe no iron limitation early in the season, which is at odds with this explanation. If Fe was not limiting it should either accumulate in the dissolved phase, be scavenged, or taken up by the phytoplankton and stored even though it is not liming. A re-phrase here might be necessary.

Lines 383-384: 'The short transient periods of increased wind stress thus appear to provide temporal relief from Fe stress'. Where is the data (wind, mixed layer depths, dFe concentrations, Fe stress status) to support this?

Lines 389: Increased nitrate concentrations throughout the growing season: do the authors need to invoke iron limitation as reducing the availability of photosynthetic reductant for nitrate reduction? The increase in nitrate concentration is large  $\sim$ 8uM: could physical process, such as a greater contribution of more recently upwelled water, be used to explain this? Or is the temperature-derived ML not capturing enhanced surface stratification later in the season that restricts downward mixing of more nitrate-depleted surface-most waters down to the incubation water collection depth? To test this the authors could calculate the buoyancy frequency in addition to the mixed layer depth.

Line 406-407: In the high latitude of North Atlantic and potentially North Pacific the cryosphere is important to seasonal dynamics? (ice/ground melting leading to enhanced stratification etc.)

---

## Referee Comment (RC2) · Anonymous Referee #2 · 24 Apr 2018

general comments: In this paper by Ran-Keogh et al the authors present data on nutrient (iron) addition bioassay style experiments conducted in the sub-Antarctic zone of the S. Ocean. The papers describes the varying response to iron–addition on the phytoplankton community over the growing season and characterises changes in physiology, nutrient uptake and community composition. The paper then discusses potential causes of the relationship between biological demand for Fe and supply. The paper is well presented and is a useful addition to the important understand of the controls and limitations on primary production in this important oceanographic region. My main concern is that the authors are too strong in their conclusions (especially relating to the seasonal cycle) from a limited dataset and sections of the paper should better reflect these limitations of the study.

specific comments: Three incubation experiments have been conducted. Which this is valuable data it is still only three data points throughout the growing season. As such conclusions as to how this data relates to a seasonal cycle should be stated with a bit more consideration. Especially as often the authors claim a development in iron-stress over the growing season while the most iron-stressed community seems to be mid-season? While this could be due to the selection of for example cells with reduced iron-requirements as iron-limitation develops it needs more openly discussed.

The authors infer accumulation of detached chlorophyll-binding protein as a mechanisms for low Fv/Fm during iron stress. If this is the case why does Fv/Fm not reduce in the set-up conditions (table 1). This is potentially a change in community – although the authors suggest the community is pretty consistent. Can the authors give a reason for this. Possibly plot Fm:Chl or similar to help support the arguments?

Is there a water temperature effect on magnitude of the deltaFv/Fm throughout similar studies from the author?

technical corrections Line 27 – suggests depend is greater than supply – the supply rate could still be high Line 74 – include a reference for this statement Line 87 – define more what eddy-strom interactions mean Line 232 – Be clear if there was no sig difference throughput the experiment Line 235 - I think the sigmaPSII data is in supplementary but please refer to this in the text Figure 2 – I think the lines are mis-labelled - +Fe and control should be the other way around? Line 304 - can you reference a paper that shows or discussed this bottle effect in more detail

---

## Author Comment (AC1) · 18 May 2018

This manuscript presents an attempt to evaluate the seasonal cycle of iron stress in the sub-Antarctic zone using three bottle-scale iron enrichment experiments conducted in December–February. The novel aspect of the study is the reoccupation of the experimental site over 2 months. Whilst these observations cannot, by some margin, be used to confidently state overarching changes in SAZ iron stress in these months, they are still valuable to the scientific community and worthy of publication in Biogeosciences. I however have a number of comments that should be addressed prior to publication. In particular I think the authors should more carefully/critically evaluate how far their experiments can actually be used to evaluate the seasonal development of Fe limitation in the sub-Antarctic zone without an analysis of supporting depth-resolved Fe, mixed layer depths, and PAR data. Upon reflection of the former, some rephrasing of the manuscript is required. Some additional important method details are also lacking.

> We agree with the reviewer that some of the conclusions were too strong for what the data presented could tell us in regards to iron supply mechanisms across the seasonal cycle. Specific sections in the text have been amended to now refer to potential phytoplankton growth and productivity.

> Some examples of how the conclusions have been adjusted are provided below.

> Line 25: "Here we demonstrate that at the beginning of the growing season, there is sufficient iron to meet the demands of the phytoplankton community, but that as the growing season develops the mean iron concentrations in the mixed layer decrease and are insufficient to meet biological demand."

> Line 434: "Irrespective of the different supply mechanisms; winter-entrainment, storm driven entrainment, diapycnal diffusion, photochemical reduction or microbial regeneration, the iron supply to the mixed layer is not sufficient for phytoplankton to reach to reach maximum growth potential and completely drawdown all available macronutrients."

> Additionally, in table 1 depth-resolved mean concentrations of all nutrients (including DFe) have been added which show seasonal depletions across all, except for nitrate. The reasons for this are explicitly discussed below in response to specific comments.

The paper is in general well written and referenced and the figures and tables are clear and complete. My comments below are listed in order through the manuscript, not by importance.

Specific comments:

Line 18: Variability in iron supply also includes dust (not mentioned)

> Atmospheric deposition has been added as a potential source of iron supply in the abstract now.

Line 21: "The variability in iron availability is due to an interplay between winter entrainment, diapycnal diffusion, storm-driven entrainment, atmospheric deposition, iron scavenging and iron recycling processes."

Line 21–22: 'incubation experiments were used to determine the importance of (iron) supply mechanisms'. Can they really be used for this? All they actually indicate is the biological response to incubation, not sources? A thorough analysis of Fe supply and demand, in conjunction with the bioassays, would be needed to do this (only 3 Fe values are reported).

I agree that this statement is misleading, and the incubation experiments cannot determine the importance of the supply mechanisms. As such this sentence has been modified.

Line 22: "Biological observations utilising grow-out iron addition incubation experiments were performed at different stages of the seasonal cycle within the SAZ to determine whether iron availability at the time of sampling was sufficient to meet biological demands at different times of the growing season."

Line 26–27: The results presented do not support the claim of progressive Fe depletion - phytoplankton appear to respond more to the Fe later in summer, but the in-situ Fe concentrations stay the same.

Whilst the in situ concentrations at the specific experimental sampling depth do not change throughout the season, the mean values across the mixed layer and euphotic zone do change. Sections describing these changes have been added to the result and discussion where necessary, with the mean DFe concentrations in the mixed layer added to table 1. The specifics of the seasonal changes in the Fe inventory will be discussed in a companion paper (Mtshali et al., In Prep).

Line 239: "Mean silicate concentrations in the mixed layerwere considered limiting and decreased between experiments (1.49 - 0.84 µM), while phosphate and dFe also displayed a gradual seasonal depletion (0.77 - 0.65 µM and 0.22 - 0.09 nM respectively); whereas nitrate concentrations increased throughout the growing season (10.41 - 12.92 µM) (Table 1)."

Line 72–73: Whilst excess nitrate+nitrite excretion under Fe stress may play a role, I think most would argue that high rates of resupply (relative to inefficient biological removal) overwhelmingly control the elevated residual nitrate in the Southern Ocean.

I agree with the comment here, that the excess nitrate is controlled by high rates of resupply. As such, the sentence has been modified to include both statements as below.

Line 76: "The result of this is excretion of excess nitrate and nitrite back into the water column, which combined with the high rates of resupply relative to biological uptake, can culminate in HNLC conditions typical of the Southern Ocean."

Line 97: 'were' change to 'that were'?

Were has been changed to that were - see line 102.

Line 99: Do the results actually indicate a change the photosynthetic efficiency? As the authors say themselves, most of the Fv/Fm change could well be due to pigment changes that have little to do with PSII efficiency.

This sentence has now been changed to reflect more broad changes in photophysiology, with what the specific changes mean to be left until the discussion.

Line 101: "This was done through a series of ship-board grow-out nutrient addition incubation experiments that were performed to determine the extent to which the addition of iron at different times of the growing season would relieve the phytoplankton from iron limitation driving changes in photophysiology, chlorophyll-a biomass and growth potential."

Line 99: Change 'biomass' to 'chlorophyll-a biomass'?

Biomass has been changed to chlorophyll-a biomass - see line 104.

Line 113: Table 'X'?

I have moved the reference to table 1 as this could lead to confusion from the reader where table 2 has the results of the changes in biomass. Table 1 is now referred to on line 117 and its reference deleted from line 107.

Line 115: CTD abbreviation defined?

Line 120: "trace metal clean CTD (Conductivity Temperature Depth) rosette system"

Line 122: Why was water 'allowed to settle' in Go-Flo samplers? To increase the overall phytoplankton concentrations in the incubation bottles?

Apologies this section was misleading, the reason for not sampling the GoFlo bottles immediately is to allow the air circulation system of the container to filter out particles that may have entered the container when transferring the GoFlo bottles from the rosette, thereby reducing any potential contamination risks. As such I have removed this phrase from the sentence so that it now reads:

Line 127: "Water for experiments were transferred unscreened into acid-washed 50 L LDPE carboy (Thermo scientific) to ensure homogenization"

Section 2.2: - More details on the incubation experiment setup needed: What was the actual incubator? A culture cabinet? Something custom built? Please give details. - How does the PAR values supplied in the incubator differ from in-situ values? For this the authors will need to calculate an average ML PAR using their observations of ML, CTD PAR and ship-instrument PAR. Although this might be a pain, it might really help to pick apart the

difference in growth environments experienced by the community just prior to incubation, and thereby help to interpret their response to the altered conditions.

> The incubator was a modified fridge that was fitted with adjustable LED light strips with time control along with a cooling fan for temperature control, the brand of the incubator is Minus40 Specialised Refrigeration.

> Line 141: "All incubations were performed within customised Minus40 Specialised Refrigeration™ units, which were fitted with adjustable (intensity and timing) LED strips as well as a thermostat and cooling fan for temperature control."

> We thank the reviewer for their suggestions to include PAR. A new row has been added to table 1 that includes the mean ± standard deviation mixed layer PAR calculated for the day of initiation from the co-located glider deployment. The light environment in the incubator was more closely matched to the average PAR in experiments 2 and 3 (~6.5 mol photons $m^{-2}$ $d^{-1}$), as opposed to experiment 1 where it was on ~12 mol photons $m^{-2}$ $d^{-1}$ lower than the average mixed layer PAR.

The following text has been added to the discussion section to highlight in situ versus incubator PAR in the experiments:

> Line 330: "The total daily PAR in the incubators ranged from 6.52 - 6.99 mol photons $m^{-2}$ $d^{-1}$, which is in good agreement for the in situ light environments of experiments 2 and 3. However, this was a ~62% decrease in the daily PAR that the phytoplankton community in experiment 1 were previously subjected to. Such a decrease in PAR would be expected to lead to a decrease in the downregulation of PSII by photodamage, coincident with an anticipated response in community structure. This could explain the observed increase in $F_v/F_m$ and decrease in $σ_{PSII}$, as larger cells tend to have a higher $F_v/F_m$ and small $σ_{PSII}$ in comparison to smaller cells (Suggett et al., 2009). Indeed we did observe a change in the community structure for experiment 1 (Fig. S2), suggestive that a decrease in light pressure resulted community response in the control treatment. However, the lack of taxonomic data at 72 h makes it difficult to distinguish whether the primary driver of this response is physiological, taxonomic or a combination of both."

Lines 130-131: Which experiments were in duplicates and which in triplicates? Figure 2 states n=3 or n=5, so I do not understand this. Please clearly indicate number of biological replicates (number of bottles with the same treatment) and technical replicates (i.e. FRR/chl measurements made from the same bottle).

> During each experiment both treatments had 16 bottles, whilst some bottles were sampled at time points for key parameters (i.e. nutrients, chlorophyll-a, FRRf), some bottles were terminated at specific timepoints to collect large volume samples for HPLC. No technical replicates were performed on the same bottle. This has been clarified with the following additional text:

Line 135: "Experiment incubations were conducted as biological replicates with 16 bottles per treatment for each experiment, these were sub-sampled at set time points for key variables as outlined in the Supplementary Information Table S1."

A sub-sampling table has been added to supplementary information to explain in greater detail the specific sampling strategy.

| Experiment | Variable | Timepoints (h) | | | | | | |
|---|---|---|---|---|---|---|---|---|
| | | 0 | 24 | 48 | 72 | 120 | 144 | 168 |
| 1

+Fe = 16 bottles

Control = 16 bottles | FRRf | 3 | 3 | n/a | 3 | 3 | n/a | 6 |
| | Chl-a | 3 | 3 | n/a | 3 | 3 | n/a | 6 |
| | Nutrients | 3 | 3 | n/a | 3 | 3 | n/a | 6 |
| | HPLC | 3 | n/a | n/a | n/a | n/a | n/a | 3 |
| 2

+Fe = 16 Bottles

Control = 16 Bottles | FRRf | 3 | 3 | n/a | 3 | 5 | n/a | 10 |
| | Chl-a | 3 | 3 | n/a | 3 | 5 | n/a | 10 |
| | Nutrients | 3 | 3 | n/a | 3 | 3 | n/a | 7 |
| | HPLC | 3 | n/a | n/a | n/a | 2 | n/a | 3 |
| 3

+Fe = 16 Bottles | FRRf | 3 | 5 | 5 | n/a | n/a | 12 | n/a |
| | Chl-a | 3 | 5 | 5 | n/a | n/a | 12 | n/a |

| Control = 16 Bottles | Nutrients | 3 | 3 | 3 | n/a | n/a | 6 | n/a |
|---|---|---|---|---|---|---|---|---|
| | HPLC | 3 | n/a | n/a | n/a | n/a | 3 | n/a |

**Table S1: Sub-sampling strategy for biological replicates of variables measured within each experiment. The number of samples collected for each variable at each timepoint is listed, where samples that were not collected is denoted by 'n/a'.**

Line 133-134: Can this approximate time of sub-sampling be included in Table 1? This might help to interpret the results (for instance, if daytime, the Fv/Fm increase in Exp 1 due to differences in PSII damage/down regulation between days).

A new row has been added to table 1 to include the initiation times, which were 07:00 for experiment 1, 20:00 for experiment 2 and 02:00 for experiment 3. In addition, text interpreting the results with respect to PAR has been address with specific reviewer comments below.

Also see line 330 in the manuscript for the specific text that addresses the effect of PAR in interpreting the results.

Line 144: Exactly how were the dFe samples filtered (method, on-ship/land)?

The DFe samples were filtered and analysed on land. This has been clarified with the following text:

Line 154: "Dissolved iron samples (DFe) were filtered through 0.2 µm cartridge filters (Acropack) equipped with a 0.45 µm pre-filter, drawn into acid washed 125 mL LDPE bottles (Nalgene, Thermoscientific), acidified with 30% HCl suprapur to pH ~1.7 (using 2 mL L$^{-1}$ criteria), double bagged and stored at room temperature until analysis on land at the Université de Bretagne Occidentale (UBO), France using the Chemiluminescence – Flow Injection Analyser (CL-FIA) method (Obata et al., 1993; Sarthou et al., 2003)."

Line 154: Was the instrument a FastTrackaII?

The instrument was a FastOcean integrated with a FastAct laboratory system attached as indicated in text – line 166.

Line 168: Was the FastPro software used or was custom code used? If the latter please give details.

The FastPro8 software (v1.0.55) was used and clarified in the text.

Line 180: "Data from the FRRf were analysed to derive the fluorescence parameters as defined in Roháček (2002), by fitting transients to the model of Kolber et al. (1998) using the FastPro8 software (v1.0.55)."

Section 2.8: What was the test for means comparison between treatments? T-test?

The test for comparison between treatments and time was ANOVA - analysis of variance. This has been clarified in the text - Line 229.

"Sample means and standard deviations were calculated using Python, followed by tests for normality and equal variance prior to analysis of variance (ANOVA) to determine treatment effects (SciPy v0.17.1, Python v3.6). Significant results are reported at the 95% confidence level ($p < 0.05$)."

Lines 303-304: ' The rapid increase in Fv/Fm in both treatments at 24 h is likely due to bottle effects i.e. a change in light environment.' - This experiment needs discussing in more detail as the community response is clearly compatible with relief of resource limitation of larger cells. How did the light change between in-situ conditions and that in the incubator? What was the integrated PAR over the previous hours prior to the Fv/Fm measurement being made (if not at night time)? Perhaps different levels of PSII damage/down-regulation could be an explanation. Could the observed community shift not contribute to the Fv/Fm increases, i.e. do larger cells not typically have higher Fv/Fm (Sugget et al. 2009)? Should any chance of Fe contamination in the control bottles (e.g. during the 50L carboy) be acknowledged, as this would also be consistent with the observed responses?

The most likely cause of this rapid change in $F_v/F_m$ in both treatments in Experiment 1, is as the reviewer correctly suggests a result of the change in the light environment between in situ and incubator conditions. The average daily PAR for the incubators was 6.52 - 6.99 mol photons $m^{-2}$ $d^{-1}$, which was a 3% increase for experiment 2 and a 16% decrease for experiment 3, in comparison to a 62% decrease for experiment 1 when compared to average daily in situ PAR.

Contamination of the 50 L carboy is unlikely to be a source of contamination as this would have propagated to the control treatments of experiments 2 and 3.

The observed community shift could explain the increase as larger cells do tend to have a higher $F_v/F_m$ and $\sigma_{PSII}$, but without taxonomic data at 72 h when this change occurs it is impossible to determine whether this response is physiological, taxonomic or a combination.

The following text has been added to the discussion in an attempt to adequately address all of the above:

Line 330: "The total daily PAR in the incubators ranged from 6.52 - 6.99 mol photons $m^{-2}$ $d^{-1}$, which is in good agreement for the in situ light environments of experiments 2 and 3. However, this was a ~62% decrease in the daily PAR that the phytoplankton

community in experiment 1 were previously subjected to. Such a decrease in PAR would be expected to lead to a decrease in the downregulation of PSII by photodamage, coincident with an anticipated response in community structure. This could explain the observed increase in $F_v/F_m$ and decrease in $\sigma_{PSII}$, as larger cells tend to have a higher $F_v/F_m$ and small $\sigma_{PSII}$ in comparison to smaller cells (Suggett et al., 2009). Indeed we did observe a change in the community structure for experiment 1 (Fig. S2), suggestive that a decrease in light pressure resulted in a community response in the control treatment. However, the lack of taxonomic data at 72 h makes it difficult to distinguish whether the primary driver of this response is physiological, taxonomic or a combination of both."

References to seasonal Fe supply (e.g. lines 315-318, 341 onwards): Reduced Fe concentrations through the growing season are not actually observed. Furthermore, Fv/Fm values are low at both the beginning and end of the growing season. The only data the authors have to go on is the more pronounced phytoplankton responses to Fe relative to controls later in the growing season. Please rephrase these sections to more clearly indicate specifically what your bioassay results can actually say about seasonal changes in Fe supply to mixed layer waters.

Depth-resolved nutrient concentrations have been added to table 1 and discussed in the results.

Line 239: "Mean silicate concentrations in the mixed layer were considered limiting and decreased between experiments (1.49 - 0.84 µM), mean phosphate and DFe also displayed a gradual seasonal depletion (0.77 - 0.65 µM and 0.22 - 0.09 nM respectively); whereas mean nitrate concentrations increased throughout the growing season (10.41 - 12.92 µM) (Table 1)."

However, we are in agreement with the reviewer that the conclusion made based upon the data presented here are too strong. As such, specific references to iron limitation now refer to their specific effects upon maintaining potential maximal growth and productivity.

Lines 311-312: 'large diatoms would require an increased silicate concentration, which is a limiting macronutrient in this region'. Is silicate a limiting nutrient at your site? Concentrations over 1uM were measured and chlorophyll-a and diatoms were enhanced in all +Fe treatments without added silicate.

Silica limitation is a well known effect in the sub-Antarctic zone as discussed in previous studies (see Hutchins et al., 2001 & Boyd et al. 2010). Furthermore, as I have discussed the silica requirements for small diatoms is much less than that of large diatoms. So without a more in depth analysis of the community structure, i.e. microscopy samples, it is impossible to determine whether this shift seen in the treatments to more diatom dominated is large or small cells.

To clarify this further, the following statement was added to the text.

Line 345: "The addition of iron also resulted in changes at the community level switching from haptophyte to diatom dominated communities (Fig. S2) despite apparent silica limitation (1.49 - 0.84 µM), typical of the region (Hutchins et al., 2001; Boyd et al. 2010). This suggests a switch to smaller diatoms, which have lower silica requirements than larger ones (Hutchins et al., 2001), however without microscopy it is not possible to say for sure."

Line 369-373: The authors state mean chlorophyll over the euphotic zone was higher than that over that of the mixed layer and then interpret this as a result of insufficient iron within the mixed layer. Whilst this is a possibility, could accumulation of lower light acclimated (higher chlorophyll/cell) phytoplankton below the mixed layer not equally play a role?

This is definitely a possibility that phytoplankton below the mixed layer but within the euphotic zone have increased their chlorophyll:carbon ratios, so an analysis was performed to look at the backscatter (another proxy for phytoplankton biomass) and we do see increased values within this sub-mixed layer zone. Whilst there is evidence of enhanced chlorophyll:carbon ratios, this sub-mixed layer population has higher values in both parameters when compared to the mixed layer.

Sections of chlorophyll and backscatter have been added to the supplementary information to show this effect more clearly, see supplementary figure 3.

[Figure]

Line 418: "Precaution must however be taken when investigating changes in Chl-a concentration, as a proxy for phytoplankton biomass (Behrenfeld et al., 2016; Bellacicco et al., 2016; Mignot et al., 2014; Westberry et al., 2008; Westberry et al., 2016), as the higher average concentration over the euphotic zone (0.8 mg m$^{-3}$) relative to the shallower mixed layer (0.4 mg m$^{-3}$) may represent a Chl-a packaging effect due to lower light levels at depth. As such, concurrent particulate backscatter ($b_{bp}$) (Fig. S3b) was investigated as an alternate proxy for phytoplankton biomass (Loisel et al., 2002; Stramski et al., 1999), which similarly depicted the presence of a subsurface bloom in response to anticipated iron relief at depth."

Lines 374–378: The mechanism the authors describe to explain the lack of Fe stress despite low Fe concentrations is not clear. The authors state that the 'limiting' nutrient 'would be expected to be severely depleted through biological uptake regardless of resupply'. But the authors state that they observe no iron limitation early in the season, which is at odds with this explanation. If Fe was not limiting it should either accumulate in the dissolved phase, be scavenged, or taken up by the phytoplankton and stored even though it is not liming. A re-phrase here might be necessary.

This phrase is referring to the principle idea that DFe concentrations do not make a good proxy for iron limitation because it does not take into account the bioavailability of iron. There are additional sources of iron that are not taken into account with this measure, i.e. ligands, particulates etc. (and in addition the rate of supply is not measured). The results from the full depth profile confirm that iron does not accumulate in the dissolved phase (see figure below taken from Mtshali et al. (In Prep)), mean DFe concentrations decrease across the growing season to minimum concentrations in February. The mean concentrations of DFe in the mixed layer have since been added to Table 1.

[Figure]

Figure: Mean concentrations of DFe (nM) in different depth bins during the four occupations of the SAZ. 0 – 200 m (winter reservoir defined by depth of maximum MLD), 0 – 82 m (euphotic zone defined by mean 1% light depth), subsurface reservoir 82 - 200 m and the mixed layer reservoir (MLD = 190 m, 32 m, 55 m and 43 m in July, December, January and February, respectively).

Within the current data set we are unable to calculate how much DFe is lost through scavenging, but it is possible that a significant portion of iron within the surface layers may be lost to this process. If phytoplankton had taken up this iron and stored it internally for when it may become a limiting nutrient then we would not see the responses present in the experiments when provided additional iron.

To provide further clarity, the text has been amended to as follows. Line 376:

"The transition from no response in experiment 1 to an increased response in experiments 2 and 3 is indicative of an increase seasonal iron limitation, similar to that observed in the high latitude North Atlantic (Ryan-Keogh et al., 2013), where available iron is depleted early in the growing season and additional resupply is insufficient to meet biological demands during the latter parts of the growing season, driving characteristic HNLC conditions. A progressive decrease in ambient iron concentrations (mean in the mixed layer; Table 1) in the SAZ, are also suggestive of a seasonal progression of iron limitation, however worth bearing in mind is that nutrient concentrations are often a poor indicator of iron limitation, as any limiting nutrient would be expected to be severely depleted through biological uptake with resultant ambient concentrations that remain close to zero despite possible event scale supply (Ryan-Keogh et al., 2017a)."

Lines 383-384: 'The short transient periods of increased wind stress thus appear to provide temporal relief from Fe stress'. Where is the data (wind, mixed layer depths, dFe concentrations, Fe stress status) to support this?

Despite having the wind, MLD and mean in the mixed layer DFe concentrations, we agree that the dataset cannot support such a bold statement, which has thus been removed. However, we do feel that the data supplied in context with the references (Little et al., In Review) are sufficient to support the importance of sub-seasonal storm events in surface mixed layer DFe supply (e.g. periods of low wind stress lead to very shallow and persistent mixed layers with proposed DFe limitation driving subsurface blooms). The text has since been modified, line 401:

"A SAZ glider study by Little et al. (In Review) corroborated these findings with summer matchups in small-scale temporal variability (< 10 days) in wind stress, MLD and chlorophyll that emphasizes the interconnectedness between physical drivers and their biological response. Despite the similarity in the scales of variability, no correlation was observed between MLD and Chl-a, which is explained by the variable response that MLD adjustments drive, i.e. dilution (a decrease in Chl-a with increasing MLD) and growth (an increase in Chl-a with increasing MLD in response to nutrient entrainment) (Fauchereau et al., 2011). Both of these scenarios can be observed in the glider time series from this study (Fig. 4), where increased wind stress and deeper MLDs were associated with both reduced (15 – 29 December) and enhanced (29 January – 7 February) Chl-a. The mid- to late summer experiments were set up during periods of low wind stress (<0.2 N m$^{-2}$) with shallow MLDs, which may corroborate the positive response to iron relief observed in experiments 2 and 3. Worth noting is that the time period between 10 January and 29 January is when the SAZ experienced uncharacteristically low winds (Braun, 2008) for an extended period of time, driving shallow MLDs (~20 m) and the development of subsurface Chl-a (Fig. S3a), indicative of iron limitation within the mixed layer and a supply mechanism (seasonal/sub-seasonal/remineralized or storm driven) that is not sufficient to meet mixed layer phytoplankton demands. Precaution must however be taken when investigating Chl-a concentration as a proxy for phytoplankton biomass (Behrenfeld et al., 2016; Bellacicco et al., 2016; Mignot et al., 2014; Westberry et al., 2008; Westberry et al., 2016), as a higher average concentration over the euphotic zone (0.8 mg m$^{-3}$) relative to the shallower mixed layer (0.4 mg m$^{-3}$) may represent a Chl-a packaging effect due to lower light levels at depth (rather than an increase in biomass). As such, particulate backscatter ($b_{bp}$) (Fig. S3b) was investigated as an alternate proxy for phytoplankton biomass (Loisel et al., 2002; Stramski et al., 1999), which similarly depicted the presence of a subsurface bloom in response to anticipated iron relief at depth."

Lines 389: Increased nitrate concentrations throughout the growing season: do the authors need to invoke iron limitation as reducing the availability of photosynthetic reductant for nitrate reduction? The increase in nitrate concentration is large 8uM: could physical process, such as a greater contribution of more recently upwelled water, be used to explain this? Or is the temperature-derived ML not capturing enhanced surface stratification later in the season that restricts downward mixing of more nitrate depleted surface-most waters down to the incubation water collection depth? To test this the authors could calculate the buoyancy frequency in addition to the mixed layer depth.

Please note that an analysis of replicate macronutrient samples (DIN + phosphate) were performed due to quality controls found within a concomitant study (Mtshali et al, In Prep) and therefore the initial conditions have been updated to reflect this. In particular note that there is still an increase in nitrate but this is not as extreme as 8 uM, the increase is now ~3 uM.

The buoyancy frequency was calculated and is presented in the figure below, with the experimental dates and depths plotted. Towards late summer there is the appearance of a secondary stratification layer but this is below the experimental depth. So mixing between the surface and the experimental depth is unrestricted during each of the occupations. However, the variability in the mixed layer and stratification layer could result in fluxes from below that could explain the increase in nitrate.

[Figure]

Line 427: "What is potentially hard to reconcile with sustained seasonal productivity and a seasonal decrease in phosphate, silicate, and DFe is the observed increase in nitrate. However, this too is suggestive of community level iron limitation, as iron limitation can reduce the availability of photosynthetic reductant for nitrate reduction which can lead to the excretion of excess nitrate back into the water column (Cochlan, 2008; Lucas et al., 2007; Milligan and Harrison, 2000; Moore et al., 2013; Price et al., 1994). This, together with the likely resupply of nitrate from below the mixed layer via sub-seasonal storm events, which is not accessible to phytoplankton uptake due to iron limitation of nitrate reductase, could account for the observed seasonal increase in mixed layer nitrate."

Line 406-407: In the high latitude of North Atlantic and potentially North Pacific the cryosphere is important to seasonal dynamics? (ice/ground melting leading to enhanced stratification etc.)

The cryosphere is important in these other high latitude regions, so the sentence has now been amended to the following:

Line 455: "The biogeochemical significance of the Southern Ocean, including the highly productive Atlantic sector, will increase with respect to climate change

(Marinov et al., 2006); particularly as the Southern Ocean is a HNLC region where the cryosphere is critical to seasonal dynamics (Massom and Stammerjohn, 2010)."

---

## Author Comment (AC2) · 18 May 2018

General comments: In this paper by Ryan-Keogh et al the authors present data on nutrient (iron) addition bioassay style experiments conducted in the sub-Antarctic zone of the S. Ocean. The papers describes the varying response to iron–addition on the phytoplankton community over the growing season and characterises changes in physiology, nutrient uptake and community composition. The paper then discusses potential causes of the relationship between biological demand for Fe and supply. The paper is well presented and is a useful addition to the important understand of the controls and limitations on primary production in this important oceanographic region. My main concern is that the authors are too strong in their conclusions (especially relating to the seasonal cycle) from a limited dataset and sections of the paper should better reflect these limitations of the study.

Specific comments:

Three incubation experiments have been conducted. Which this is valuable data it is still only three data points throughout the growing season. As such conclusions as to how this data relates to a seasonal cycle should be stated with a bit more consideration. Especially as often the authors claim a development in iron stress over the growing season while the most iron-stressed community seems to be mid-season? While this could be due to the selection of for example cells with reduced iron-requirements as iron-limitation develops it needs more openly discussed.

> The most iron stressed community is mid-season, if examining the photophysiology alone, however experiment 3 displayed the greatest increases in growth rates following iron addition. A statement to this effect has been added to the discussion.

> Line 412: "When examining the photophysiology alone, experiment 2 displayed the greatest response to iron addition (Fig. 2c) with significant responses also observed in Chl-a derived net growth rates (Fig. S1) and nitrate drawdown rates (Table 2). Experiment 3 displayed the greatest increases in growth rates following Fe addition (Fig. S1), while significantly higher $F_v/F_m$ was similarly observed (Fig. 2e)."

> We agree with reviewer 2, who similarly raised concerns like reviewer 1, that more consideration should be taken in regards to the conclusions. As such, the text has been modified to discuss the implications of iron limitation here on the potential maximal growth rates and productivity.

> Some examples of how the conclusions have been adjusted are provided below.

> Line 25: "Here we demonstrate that at the beginning of the growing season, there is sufficient iron to meet the demands of the phytoplankton community, but that as the growing season develops the mean iron concentrations in the mixed layer decrease and are insufficient to meet biological demand."

> Line 434: "Irrespective of the different supply mechanisms; winter-entrainment, storm driven entrainment, diapycnal diffusion, photochemical reduction or microbial regeneration, the iron supply to the mixed layer is not sufficient for phytoplankton to

reach to reach maximum growth potential and completely drawdown all available macronutrients."

The authors infer accumulation of detached chlorophyll-binding protein as a mechanisms for low Fv/Fm during iron stress. If this is the case why does Fv/Fm not reduce in the set-up conditions (table 1). This is potentially a change in community – although the authors suggest the community is pretty consistent. Can the authors give a reason for this. Possibly plot Fm:Chl or similar to help support the arguments? Is there a water temperature effect on magnitude of the deltaFv/Fm throughout similar studies from the author?

One potential reason for the $F_v/F_m$ not reducing during the set-up conditions is that the phytoplankton species (haptophyte dominated - see Fig. S2) are living under steady-state iron limited conditions during experiments 2 and 3, high values of $F_v/F_m$ have previously been observed under steady state iron limitation in culture (Parkhill et al. 2001; Price 2005). However, with the lack of sufficient community structure data it is hard to determine which signal is dominating the measurements presented here. As the experiments will display a physiological signal (i.e. nutrient stress), superimposed with a taxonomic signal (i.e. different phytoplankton groups have different baseline $F_v/F_m$). It is also not possible to rule out the effects of light intensity on suppressing the initial $F_v/F_m$ measured, through the downregulation of PSII, when setting up the experiments. During the cruise, a CTD was deployed at 03:00 local time before sunrise, and a depth profile of FRRf was collected. Samples collected at 10m and 50m indicated a much higher $F_v/F_m$, with 0.32 and 0.39 respectively, in comparison to the initial samples collected 4 hours later after sunrise. Potential reasons for this discrepancy is that the dark acclimation step may have not fully relaxed the initial samples before measurement. Indeed, one of the results of the incubation was a ~62% decrease in the light exposure that the experimental bottles received, which potentially explains why both the controls and Fe increase their $F_v/F_m$ by ~0.15.

[Figure]

Depth profile of $F_v/F_m$ of the same station for samples that were collected on a CTD cast 4 hours prior to the experimental CTD cast.

Temperature was examined as a potential driver of $\Delta(F_v/F_m)$ during previous studies as part of my PhD, but was not found to be a significant driver in these studies and therefore excluded from the analysis (Ryan-Keogh et al., 2013; Ryan-Keogh et al., 2017). I have attached here 2 figures of data from these studies (unpublished), showing $\Delta(F_v/F_m)$ against temperature. However, given the small temperature range in the experimental set up, 10.44 - 10.8, it is unlikely that there would be any temperature effect in dictating the range of $\Delta(F_v/F_m)$ values measured in the experiments.

[Figure]

HLNA results from Ryan-Keogh et al. 2013

[Figure]

Ross Sea results from Ryan-Keogh et al. 2017

Displayed here is the full time series of the changes in $F_m$ Chl$^{-1}$ and $F_v$ Chl$^{-1}$ from each experiment, evident in panels c & e is that the iron addition creates a difference in $F_m$ Chl$^{-1}$ between the treatments; there is no evidence of changes in $F_v$ Chl$^{-1}$ (panels d & f). However, I feel that the information in this figure is already displayed in figure 3c and 3d.

[Figure]

Technical corrections:

Line 27 – suggests depend is greater than supply – the supply rate could still be high

> This sentence has now been modified to:

> Line 31: "suggestive of seasonal iron depletion and an insufficient resupply of iron to meet biological demand."

Line 74 – include a reference for this statement

> The following reference has been added to the text on line 80 and included in the references.

> Boyer, T. P., Antonov, J. I., Baranova, O. K., Coleman, C., Garcia, H. E., Grodsky, A., Johnson, D. R., Locarnini, R. A., Mishonov, A. V., and O'Brien, T. D.: World Ocean Database 2013, NOAA Printing Office, Silver Spring, MD, 2013.

Line 87 – define more what eddy-strom interactions mean

> This section has now been updated to only discuss the effects of storm on shear mixing, rather than the effects of eddy-storm interactions on 3D mixing. The

complexities of these different mixing mechanisms is beyond the scope of this paper and is discussed in greater detail elsewhere. See line 93 for clarification.

Line 232 – Be clear if there was no sig difference throughout the experiment

There was no significant difference at any timepoint during the experiment, the text has been updated to state this more clearly.

Line 254: "Statistical analysis confirmed that there were no significant differences in $F_v/F_m$ or chlorophyll throughout the experiment."

Line 235 – I think the sigmaPSII data is in supplementary but please refer to this in the text

References to supplementary figure S1 which shows $\sigma_{PSII}$ has been added to the text on lines 256, 264 and 272.

Figure 2 – I think the lines are mis-labelled - +Fe and control should be the other way around?

The labelling is correct on Figure 2, open circles which show the lowest values of $F_v/F_m$ and chlorophyll are the controls. The closed circles represent the +Fe treatment which has the higher values of chlorophyll and $F_v/F_m$.

Line 304 – can you reference a paper that shows or discussed this bottle effect in more detail

Additional references have now been added - see line 327. This section discussing the bottle effects evident here have also been greatly expanded and discussed in detail.

"The rapid increase in $F_v/F_m$ in both treatments from 24 h onwards is likely due to potential bottle effects i.e. a change in the light environment (Martin and Fitzwater, 1988; de Baar et al., 1990; Coale 1991; de Baar et al., 2005). The total daily PAR in the incubators ranged from 6.52 - 6.99 mol photons $m^{-2}$ $d^{-1}$, which is in good agreement for the in situ light environments of experiments 2 and 3. However, this was a ~62% decrease in the daily PAR that the phytoplankton community in experiment 1 were previously subjected to. Such a decrease in PAR would be expected to lead to a decrease in the downregulation of PSII by photodamage, coincident with an anticipated response in community structure. This could explain the observed increase in $F_v/F_m$ and decrease in $\sigma_{PSII}$, as larger cells tend to have a higher $F_v/F_m$ and small $\sigma_{PSII}$ in comparison to smaller cells (Suggett et al., 2009). Indeed, we did observe a change in the community structure for experiment 1 (Fig. S2), suggestive that a decrease in light pressure resulted in a community response in the control treatment. However, the lack of taxonomic data at 72 h makes it difficult to distinguish whether the primary driver of this response is physiological, taxonomic or a combination of both."